# Evaluation of Ocular Diameter Parameters Using Swept-Source Optical Coherence Tomography

**DOI:** 10.3390/medicina59050899

**Published:** 2023-05-08

**Authors:** Jing Dong, Jinhan Yao, Shuimiao Chang, Piotr Kanclerz, Ramin Khoramnia, Xiaogang Wang

**Affiliations:** 1Department of Ophthalmology, First Hospital of Shanxi Medical University, Taiyuan 030001, China; 2Shanxi Eye Hospital Affiliated to Shanxi Medical University, Taiyuan 030002, China; 3Hygeia Clinic, 80-286 Gdańsk, Poland; 4Helsinki Retina Research Group, University of Helsinki, 00014 Helsinki, Finland; 5The David J. Apple International Laboratory for Ocular Pathology, Department of Ophthalmology, University of Heidelberg, 69120 Heidelberg, Germany

**Keywords:** angle-to-angle, anterior chamber intraocular lens, sclera spur-to-sclera spur, swept-source optical coherence tomography, white-to-white, implantable collamer lens

## Abstract

*Purpose:* To investigate the iridocorneal angle-to-angle (ATA), sclera spur-to-sclera spur (STS), and white-to-white (WTW) ocular diameters and their potential influence on anterior chamber intraocular lens (ACIOL) and implantable collamer lens (ICL) sizing in Chinese subjects by using a swept-source optical coherence tomography system (SS-OCT). *Design:* A retrospective, observational, cross-sectional study. *Methods:* In 60 right eyes (60 subjects), the ATA, STS, and WTW were measured in six axes (0°–180°, 30°–210°, 60°–240°, 90°–270°, 120°–300°, and 150°–330°) using SS-OCT. The ACIOL and ICL sizes were calculated based on horizontal and vertical axes anterior segment data. A paired sample t-test was used to test the differences in each parameter across the six axes, the potential difference between each pair of parameters in a given axis, and the artificial lens size difference between the horizontal and vertical directions. Pearson’s correlation analysis was used to determine the potential correlation between age and AL, WTW, STS, and ATA distances. *Results:* ATA and STS were the longest on the vertical and shortest on the horizontal axis, while WTW was similar on both axes. These three parameters differed only in the vertical axis (F = 4.910, *p* = 0.008). ATA and STS were by 0.23 ± 0.08 mm (*p* = 0.005) and 0.21 ± 0.08 mm wider (*p* = 0.010) than WTW, respectively. ICL size was 0.27 ± 0.23 mm smaller when based on the horizontal than on the vertical axis parameters (*p* < 0.001), while ACIOL remained similar (*p* = 0.709). Age correlated negatively and axial length positively with all measured values. ATA, STS, and WTW correlated positively in the same axis (all *p* < 0.001). *Conclusions:* ATA and STS were longer in the vertical than in the horizontal direction, while WTW measurements remained similar. ATA and STS diameters more accurately depicted anatomic relationships for phakic IOL sizing than WTW.

## 1. Introduction

The surgical correction of refractive eye status with the absence of adequate capsular or zonular support is challenging; in eyes with a deep anterior chamber, an appropriately sized anterior chamber intraocular lens (ACIOL) could be a suitable treatment option [1,2]. Implantable collamer lenses (ICLs), as a biocompatible material of collagen and polymer, can treat a wide range of ametropias; for high myopia, they were shown to be more effective than laser in situ keratomileusis [3]. However, complications related to a nonideal postoperative ICL vault, caused by improper ICL size choices, such as glaucoma, cataract, and even crystalline lens dislocation, have been reported [4,5,6,7,8,9].

The precise measurement of anterior segment distances, including iridocorneal angle-to-angle (ATA), scleral spur-to-scleral spur (STS), white-to-white (WTW), and ciliary sulcus-to-ciliary sulcus, is important in the above-mentioned artificial lens size calculations [10,11,12]. Several methods, such as corneal topography, ocular biometry, surgical caliper, ultrasound biomicroscopy (UBM), and optical coherence tomography (OCT), have been used to measure the parameters mentioned above in the clinic. Most surgeons have relied on horizontal anterior segment distances for artificial lens size calculation [13,14,15,16]. With improved scanning speed, high resolution, fewer artifacts, and less influence from eye movements, anterior segment swept-source OCT (SS-OCT) has been used for artificial lens size calculation in the clinic [17,18].

The ATA and STS demonstrated radial variation along the vertical ellipse based on a study by Montés-Micó et al. using SS-OCT [13]. This indicated the need to verify whether there is also radial variation in the WTW [2,10]. Moreover, there is a need to confirm whether there is a potential difference between phakic IOL size determined using horizontal and vertical measurements.

Therefore, this study investigated the potential WTW radial variation along six meridians using SS-OCT and sought to find the corresponding differences in phakic IOL size determined using horizontal and vertical measurements obtained using SS-OCT.

## 2. Methods

### Subjects

This study was performed at the Shanxi Eye Hospital, Affiliated with Shanxi Medical University. The research protocol was approved by the institutional review board of Shanxi Eye Hospital, Affiliated with Shanxi Medical University (No. 2019LL130), and was conducted according to the tenets of the Declaration of Helsinki. Written informed consent was obtained from each subject. This observational study has been registered online (at the International Standard Randomized Controlled Trials at http://www.controlled-trials.com, accessed on 8 November 2021) with the registration number: ISRCTN13860301.

Consecutive cataract department outpatient subjects were enrolled between March 2021 and September 2021. Inclusion criteria were described as follows: no pathological alteration of the anterior segment (such as anterior segment uveitis, keratoconus, zonular dialysis, pseudoexfoliation syndrome, fibrosis, or scarring of the cornea), no retinal diseases impairing visual function (such as retinal detachment, macular hole, or macular edema), no previous anterior or posterior segment surgical history, best-corrected visual acuity better than 20/40, and normal intraocular pressure (within the range of 10–21 mmHg using noncontact tonometer). Patients also had to be able to cooperate with all eye examinations, such as slit-lamp examination, intraocular pressure measurement, and SS-OCT data capture. Patients who could not cooperate with the data-capturing procedure and failed to pass the image quality check were excluded.

## 3. Data Acquisition

All patients underwent SS-OCT examinations without the use of any eye drops. The axial length (AL) was measured using the Cataract App of the ANTERION SS-OCT system (software version 1.3.4.0, Heidelberg Engineering GmbH, Heidelberg, Germany). The Metrics App was used in the capture and analysis of cross-sectional anterior segment imaging in six axes. This OCT system was operated using a 1300 nm wavelength light source, which provided an axial depth of 14 mm and lateral width of 16.5 mm. It had a scan speed of 50,000 A-scans/second, with an in-tissue resolution of approximately 10 and 30 μm for axial and lateral scanning images, respectively. An experienced technician captured all the images and checked the image quality in this study. A second experienced technician rechecked the image quality. The eye-tracking function was active throughout the image-capturing procedure and captured images with good quality and confirmed the consistency by both the device and expert for data analysis. The anterior chamber width (equivalent to STS) and crystalline lens rise (the distance between the anterior crystalline lens surface and ATA line) can also be provided for ICL size calculation in this scanning model [17].

## 4. ATA, STS, and WTW Parameter Measurements

The quantitative parameters included ATA, STS, and WTW (Figure 1). The ATA was defined as the distance between the bilateral angle recess point (the intersection points between the cornea and the iris) on the same scanning-axis image. It was automatically calculated after being manually labeled. The STS was defined as the distance between the bilateral scleral spurs on the same scanning-axis image, which was automatically calculated after manually marking the anatomical position of the scleral spur. The WTW was defined as the distance between the bilateral corneal limbus (the transition between the cornea and sclera) on the same scanning-axis image, which was manually calculated using Image J software (Java version 1.8.0_172). Finally, the crystalline lens rise used for ICL size calculation was defined as the distance between the anterior crystalline lens surface and the ATA line. All manual label points and measurements, confirmed and verified by the same two experienced technicians as cited above, were collected for the final data analysis.

## 5. ACIOL and ICL Size Calculations

The ACIOL size choice was determined using the online size chart for Alcon lenses (model MTA2-7U0), which is the WTW distance with 1 mm added (Kourtney Houser. Anterior Chamber Intraocular Lens, AAO EyeWiki, Updated 13 January 2022, accessed on 2 April 2022. https://eyewiki.aao.org/Anterior_Chamber_Intraocular_Lenses). The ICL size was calculated using the NK formula version 2 (ICL size (mm, in a balanced salt solution) = 4.575 + 0.688 × (anterior chamber width with mm unit) + 0.388 × (crystalline lens rise with mm unit)) from Nakamura et al. [17] The horizontal and vertical axes anterior scanning biometric data were used for corresponding ACIOL and ICL sizing calculations, respectively.

## 6. Statistical Analysis

Statistical analyses were performed with commercial software (SPSS ver. 22.0; IBM SPSS Inc., Armonk, NY, USA) and (MedCalc ver. 12.7.0.0; MedCalc Software Ltd., Oostende, Belgium). Normality distribution was testified using the Shapiro–Wilk test. Analysis of variance was used to compare the potential differences in each direction among the three parameters. A paired sample *t*-test was used to test the differences in each parameter across the six axes, the potential difference between each pair of parameters in a given axis, and the artificial lens size difference between the horizontal and vertical directions. Pearson’s correlation analysis was used to determine the potential correlation between age and AL, WTW, STS, and ATA distances. The significance level was set at *p* < 0.05. Bonferroni-corrected *p* values (0.025) were used for multiple comparisons among the three parameters. 

## 7. Results

Initially, 85 right eyes of 85 participants were enrolled in this study. Five eyes were excluded because of unqualified image quality, and twenty eyes were excluded because the scleral spur was not visible for human labeling in all six scanning axes. Finally, 60 right eyes of 60 subjects were included in the data analysis. The demographic information and ATA, STS, and WTW measurements are summarized in Table 1.

In comparing the ATA across the six axes, the ATA in the vertical axis was the longest, and the ATA measured in the horizontal, 30°–210°, and 150°–330° axes were relatively shorter. This demonstrated that the 360° outline of the ATA forms a vertical ellipse. Moreover, the ATA difference between the vertical and horizontal directions was 0.36 ± 0.28 mm (*p* < 0.001). STS measured on the vertical axis was the longest, and that measured on the horizontal axis was the shortest, demonstrating a similar vertical ellipse outline as that of ATA. The difference in STS between the vertical and horizontal directions was 0.35 ± 0.31 mm (*p* < 0.001). WTW showed no significant difference between the vertical and horizontal axes (mean difference −0.001 ± 0.33, *p* = 0.988), indicating that the 360° outline of WTW was nearly circular (Figure 2).

A significant difference was found among the three parameters in the vertical axis (F = 4.910, *p* = 0.008) but not in the other five axes (all *p* > 0.130). For the vertical axis, ATA was 0.23 ± 0.08 mm wider than WTW (*p* = 0.005), while STS was 0.21 ± 0.08 mm wider than WTW (*p* = 0.010), and no statistically significant difference was found between ATA and STS (*p* = 0.814). 

For the ACIOL size, no significant difference was found between the horizontal and vertical sizes (12.43 ± 0.46 mm and 12.42 ± 0.38 mm, *p* = 0.709). For ICL size comparison, the horizontal ICL size was 0.27 ± 0.23 mm smaller than the vertical ICL size (*p* < 0.001). 

Age was negatively correlated with all ATA (r ranges from −0.344 to −0.477, all *p* < 0.004), all STS (r ranges from −0.308 to −0.389, all *p* < 0.017), and some WTW values (r ranges from −0.259 to −0.409, all *p* < 0.046 except vertical axis (r = −0.177, *p* = 0.175) and 120°–300° (r = −0.195, *p* = 0.136)). In contrast, AL was positively correlated with all three measured values (r ranges from 0.355 to 0.564, all *p* < 0.005; Figure 3). ATA, STS, and WTW were significantly positively correlated in the same axis (Pearson’s r values ranged from 0.703 to 0.938, all *p* < 0.001). 

## 8. Discussion

In this study, we found that the ATA and STS outlines formed vertical ellipses, whereas WTW showed a nearly circular contour in Chinese subjects, based on SS-OCT measurements. These results clearly explain what was reported in clinical studies; rotating a nontoric ICL is a simple technique to deal with oversized ICLs. They also underline that ATA and STS are more appropriate for ICL size estimation than WTW [19,20].

ACIOL size based on horizontal and vertical measurements was not markedly different, whereas the ICL size predicted from vertical measurements was more significant than that based on horizontal axis measurements. Our results also indicate a potential correlation among ATA, STS, and WTW. Moreover, age and AL also influence these parameters, which should be kept in mind when using these three parameters in a clinical context.

The scleral spur visibility rate was approximately 71% (in 60 out of 85 eyes) in this study, which was similar to that found in a previous study using time-domain AS-OCT images (1439/2008 images = 71.7%) from 502 right eyes (4 scanning images per eye) [21]. This detection rate was not markedly improved even with SS-OCT and its relatively higher image resolution. We consider that this phenomenon may have contributed to eyelid manipulation, anatomical variation, and optical correction factors for this peripheral area. 

As in the study by Montés-Micó et al., ANTERION SS-OCT was used to evaluate differences in the three parameters in the present study [13]. However, all corresponding mean values of ATA, STS, and WTW were smaller than in their study. This difference may be attributable to the differences in the ages of the study populations and the potential racial or ethnic differences. Our findings confirm a negative correlation between age and these three parameters [22,23]. Moreover, our results of the correlations of these parameters were similar to those of previous studies [10,13,24,25]. Previous studies found that the correlation between AL and WTW was not immutable; the WTW increased when the AL ranged from 24.5 to 26.0 mm, but decreased when the AL increased beyond 26 mm [22]. Therefore, a larger sample size and AL range are needed to confirm the actual correlation tendency.

Most previous studies confirmed a vertical ellipse contour for ATA, even when using different imaging technologies and ranges (vertical distance minus horizontal distance) from 0.1 mm to 0.45 mm [25,26,27,28]. Our mean difference was 0.36 mm, which fell within this range. One exception was the study Nemeth et al., which found that the mean horizontal ATA was approximately 0.71 mm greater than the vertical value when using AS-OCT [24]. We suggest that this unusual finding should be reconfirmed: scanning location may fluctuate due to eye movement when using the relatively low scanning speed of time-domain AS-OCT.

Similar to some previous studies, the vertical oval outline of STS was also confirmed by using SS-OCT in the present study. The mean difference was 0.35 mm, which was also within the range of previously published data (0.24–1.32 mm) [2,13].

Usually, the corneal diameter (regarded as the limbal distance or WTW distance) was a horizontal oval [29]. However, this was different from our findings, which demonstrated a circular WTW outline, similar to that reported by Abass et al. [30]. We speculated that the reasons may be the following. First, there may be a relatively poor agreement among devices, as the manual or automated anatomical location of WTW varied among different imaging methods [31,32]. Second, the limbus boundary, where the cornea ends and the sclera begins, is a less-defined area. Thus, it is difficult to define the location precisely in captured images, especially in the superior and inferior quadrants or eyes with unclear margins, such as pterygium, corneal ulcer, and corneal edema. Moreover, WTW was calculated from the infrared camera image, but the STS and ATA were assessed from a cross-sectional OCT image. The different sets of images, such as the image dewarping, should also be considered in a future study. Therefore, finding a relatively more stable anatomical point in OCT or ultrasound biomicroscopy images would be better.

With the optimized ICL formula, such as the NK formula version 2, the use of STS and crystalline lens rise distance as the main weight factors demonstrated excellent results for selecting the proper ICL size [17]. The predicted ICL size based on the vertical direction was 0.27 mm larger than that based on the horizontal direction when using the optimized formula in this study. This finding was consistent with the vertical oval shape of the ciliary sulcus confirmed by UBM and indicated that the vertical direction might be more suitable for ICL size calculation and more stable for ICL placement in surgery [2,28]. Moreover, if the postoperative ICL vault was less than the minimum threshold (250 μm) in the horizontal position, the surgeon should consider the ICL automatically rotating possibility from horizontal to vertical, which may result in a much lower postoperative ICL vault (<250 μm). However, if the ICL vault exceeds the maximum threshold (1000 μm) in the horizontal position, a manual rotation to the vertical direction may allow the surgeon to observe but not exchange the ICL size to achieve a better ICL vault [19]. The surgeon should also pay attention to the degree of superior anterior chamber angle changes if the ICL is implanted vertically [33].

Unlike the ICL, the ACIOL size was similar in vertical and horizontal directions. However, because the normal asymmetric anterior chamber angle distribution and superior angle were narrower, we recommend choosing a horizontal direction as the ACIOL target position [33].

Our study had several limitations. First, because of blocking by the iris tissue, the current SS-OCT system cannot clearly demonstrate the ciliary sulcus anatomy. Therefore, no sulcus-to-sulcus distance was evaluated in this study. Even if most surgeons favor ciliary sulcus distance as the standard for ICL size selection, we still believe that for some cases, the ICL haptic may not be in the exact sulcus location; this also should be noticed when we consider the ICL size selection and the most influencing calculation parameters. Second, no gonioscopic angle status analysis was included in the current study. Our two experienced technicians as cited above rechecked each image to verify the angle recess point and SS location to minimize this impact. Third, scanning patterns are currently limited to six axes, but these data may not provide detailed information in all 360° directions. This limitation can be addressed if the user can customize the scanning lines.

In conclusion, we showed that, based on SS-OCT measurements, the ATA and STS follow a vertical ellipse contour, while the WTW follows an essentially circular outline. Therefore, these features should be considered when calculating phakic IOL sizes.

## Figures and Tables

**Figure 1 medicina-59-00899-f001:**
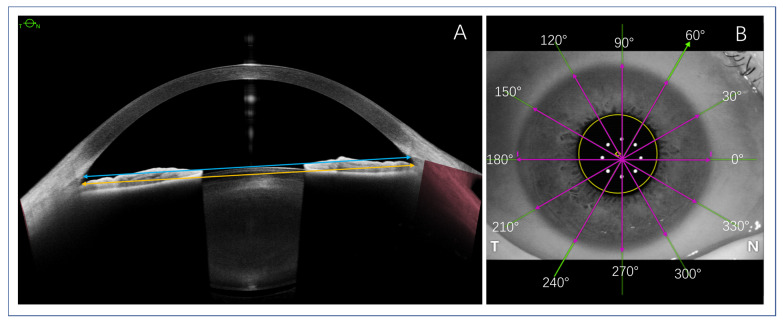
Schematic diagram of the three parameters measured. (**A**) The STS (blue double arrows solid line) and ATA (yellow double arrows solid line) measurement of 0°–180° scanning direction; Panel (**B**) demonstrated the 6 scanning directions (green lines) with 30° interval, and the overlapping pink double arrows solid lines showed the WTW distance on each scanning direction. Note: STS, sclera spur-to-sclera spur; ATA, iridocorneal angle-to-angle; WTW, white-to-white.

**Figure 2 medicina-59-00899-f002:**
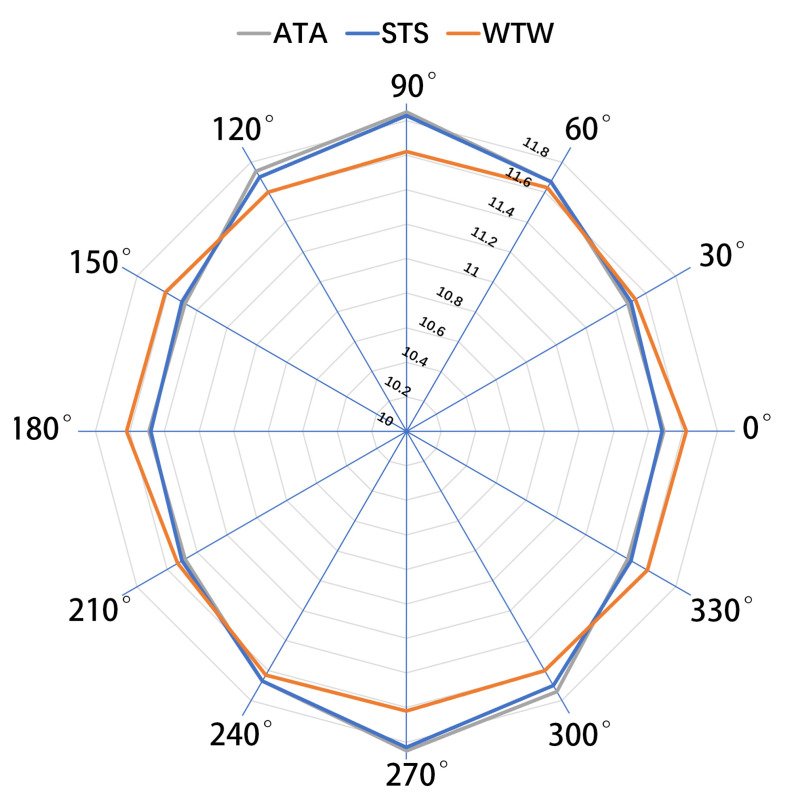
The 360° outline of ATA, STS, and WTW for the whole eye. The ATA (gray line) and STS (blue line) contour followed a vertical ellipse shape, and the WTW (orange line) contour was almost circular. Note that the radial axis begins at a 10 mm diameter. STS, sclera spur-to-sclera spur; ATA, iridocorneal angle-to-angle; WTW, white-to-white.

**Figure 3 medicina-59-00899-f003:**
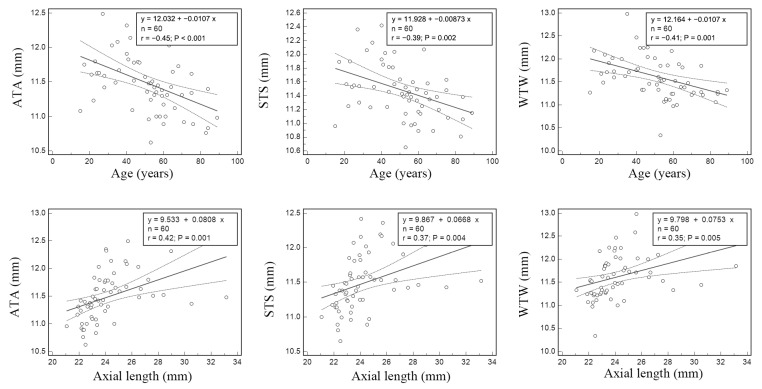
Scatter plots and regression equations. The regressions correspond to age (**top row**) or axial length (**bottom row**), and iridocorneal angle-to-angle (ATA), scleral spur-to-scleral spur (STS), or white-to-white (WTW) in the 0°–180° direction. The 95% confidence intervals for the regression curve are shown as dashed lines. The value for the Pearson’s correlation coefficient and the *p* value are shown on the inset (top right corner) of each individual plot.

**Table 1 medicina-59-00899-t001:** Demographic information and analyzed parameters (*n* = 60).

	Mean ± SD	Range
Age (years)	51 ± 18	15–89
AL (mm)	24.19 ± 2.20	21.05–33.16
CCT (μm)	522 ± 34	447–589
ACD (mm)	3.39 ± 0.43	1.99–4.20
LT (mm)	4.14 ± 0.49	2.90–5.29
ATA 90°–270° (mm)	11.85 ± 0.48	10.61–12.93
ATA 60°–240° (mm)	11.67 ± 0.47	10.70–12.69
ATA 30°–210° (mm)	11.48 ± 0.45	10.70–12.52
ATA 0°–180° (mm)	11.49 ± 0.43	10.62–12.49
ATA 150°–330° (mm)	11.48 ± 0.43	10.62–12.61
ATA 120°–300° (mm)	11.74 ± 0.43	10.91–12.74
STS 90°–270° (mm)	11.83 ± 0.47	10.62–12.80
STS 60°–240° (mm)	11.67 ± 0.45	10.95–12.63
STS 30°–210° (mm)	11.50 ± 0.43	10.80–12.45
STS 0°–180° (mm)	11.48 ± 0.40	10.65–12.42
STS 150°–330° (mm)	11.50 ± 0.40	10.82–12.44
STS 120°–300° (mm)	11.70 ± 0.42	10.72–12.56
WTW 90°–270° (mm)	11.62 ± 0.37	10.95–12.58
WTW 60°–240° (mm)	11.63 ± 0.43	10.92–12.79
WTW 30°–210° (mm)	11.53 ± 0.44	10.40–12.81
WTW 0°–180° (mm)	11.62 ± 0.47	10.34–12.98
WTW 150°–330° (mm)	11.61 ± 0.40	10.48–12.58
WTW 120°–300° (mm)	11.60 ± 0.35	10.83–12.46

AL, axial length; CCT, central corneal thickness; ACD, anterior chamber depth; LT, lens thickness; ATA, iridocorneal angle-to-angle; STS, sclera spur-to-sclera spur; WTW, white-to-white.

## Data Availability

The data that support the findings of this study are available on reasonable request from the corresponding author.

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
