# Peer review of "Evaluation of Ocular Diameter Parameters Using Swept-Source Optical Coherence Tomography"

_medicina, 2023, doi:10.3390/medicina59050899_

Round 1
Reviewer 1 Report
The manuscript is written in proper english, the content is relevant and the analysis sound.
There are only very minor things I can offer to potentially further improve the manuscript:
- The methodology should mention the exact IOL models that were calculated for this study. ICL sizing might be different to the sizing of other posterior chamber lenses like IPCL due to rigidity of the material etc., this could be specified.
- sizing changes from the ICL OCOS system to NK formula with AT/STS would have been interesting. Ideally there would be different regressions for STS/ATA/WTW in a formula.
- STS was larger than ATA in the 150 to 330° axis, can that really be?
- ICL haptic location is not necessarily the sulcus, but also the ciliary body itself, this could be briefly mentioned in the discussion, as chosing the right location to predict the sulcus to sulcus size might not eleminate all outliers.
Author Response
Dear reviewer, thank you for your excellent comments. We made a point-by-point response to each question. Check it, please.
- The methodology should mention the exact IOL models that were calculated for this study. ICL sizing might be different to the sizing of other posterior chamber lenses like IPCL due to rigidity of the material etc., this could be specified.
ANSWER: we added the IOL model information and ICL material information in the method and introduction parts, respectively. Check it, please.
- sizing changes from the ICL OCOS system to NK formula with AT/STS would have been interesting. Ideally there would be different regressions for STS/ATA/WTW in a formula.
ANSWER: we agree with that and will do this kind of investigation and research work in our next step. I hope we still have the chance to share our results with you. Thank you.
- STS was larger than ATA in the 150 to 330° axis, can that really be?
ANSWER: Table 1 demonstrated that the STS (11.48 ± 0.43mm) and ATA (11.50 ± 0.40mm) was similar in the 150 to 330° axis. They are both shorter than the corresponding WTW (11.61 ± 0.40mm) on the same axis.
- ICL haptic location is not necessarily the sulcus, but also the ciliary body itself, this could be briefly mentioned in the discussion, as chosing the right location to predict the sulcus to sulcus size might not eleminate all outliers.
ANSWER: we added some information in the discussion part. Check it, please.
Reviewer 2 Report
This is an interesting and well written paper about imaging and measuring some anatomical landmarks that could potential play an important role in the determination of a proper IOL/ ICL lens in eyes undergoing cataract or ICL refractive surgery.
I have no specific comments or queries pertaining to this manuscript.
Author Response
Dear reviewer, thank you for your comments.
Reviewer 3 Report
The authors present an important nomogram that clinicians need to follow esp if anterior segment is planned esp ICL and also for AC-IOL, etc. They detected vertical ellipses with ATA and STS and hence ATA and STS are more appropriate to use than WTW in ICL.
Author Response

(The authors gave the same response as above.)
